# How Can Cattle Be Toilet Trained? Incorporating Reflexive Behaviours into a Behavioural Chain

**DOI:** 10.3390/ani10101889

**Published:** 2020-10-15

**Authors:** Neele Dirksen, Jan Langbein, Lars Schrader, Birger Puppe, Douglas Elliffe, Katrin Siebert, Volker Röttgen, Lindsay Matthews

**Affiliations:** 1Leibniz Institute for Farm Animal Biology (FBN), Institute of Behavioural Physiology, 18196 Dummerstorf, Germany; dirksen@fbn-dummerstorf.de (N.D.); puppe@fbn-dummerstorf.de (B.P.); siebert@fbn-dummerstorf.de (K.S.); roettgen@fbn-dummerstorf.de (V.R.); 2Friedrich-Loeffler-Institut, Institute of Animal Welfare and Animal Husbandry, 29223 Celle, Germany; lars.schrader@fli.de; 3Behavioural Sciences, Faculty of Agricultural and Environmental Science, University of Rostock, 18059 Rostock, Germany; 4School of Psychology, The University of Auckland, Auckland 1142, New Zealand; d.elliffe@auckland.ac.nz (D.E.); lindsay.matthews1@gmail.com (L.M.)

**Keywords:** toilet training, cattle, operant conditioning, reflex conditioning, behavioural chain, urination

## Abstract

**Simple Summary:**

The mixing of bovine faeces and urine leads to climate-damaging ammonia emissions. If cattle could be taught to use a latrine, this would reduce the area of emissions, the separation of excreta could easily be accomplished by mechanical means, and animal health could be improved. Attempts to train toileting in cattle have shown limited success. In children, toileting is trained mainly by (i) interrupting voiding that starts outside the toilet, then taking the child to the toilet and rewarding the resumption of excretion, or (ii) placing the child on the toilet, waiting for urination/defecation and rewarding appropriate excretory behaviour. The first method is reported to be more successful. Thus, a similar procedure was evaluated for training latrine use for urination in calves. On 95% of occasions, the calves inhibited or stopped urination when receiving a signal to move to the latrine, and on 65% of occasions, they reinitiated urination in the latrine. Furthermore, during 63% of urinations in the latrine, the calves oriented towards the reward location before any food was delivered, providing additional evidence that calves can be successfully toilet trained with food rewards.

**Abstract:**

Untrained cattle do not defecate or urinate in defined locations. The toilet training of cattle would allow urine and faeces to be separated and stored, reducing climate-damaging emissions and improving animal health. In a proof-of-concept study, we evaluated a novel protocol for toilet training in cattle. Five heifer calves (and yoked controls) were trained in the voluntary (operant) behaviours of a toileting chain. Then, reflexive urinating responses were incorporated into the chain, with toileting signalled by a tactile (vibratory) stimulus. On 95% of occasions, the calves inhibited/interrupted urination when receiving the stimulus, and on 65% of these occasions, reinitiated urination in the latrine. Furthermore, during 63% of urinations in the latrine, the calves oriented to the reward location before any food was delivered, providing additional evidence that calves can be successfully toilet trained with food rewards. Yoked controls failed to learn most of the operant elements and all the reflexive responses of toilet training. The results show that reflexive behaviours can be incorporated into voluntary toileting sequences with cattle and extend the range of species that can be toilet trained. Future refinement of the protocol to allow training under practical farm conditions offers the potential to mitigate climate damage and improve animal health.

## 1. Introduction

Worldwide, cattle contribute approximately 10% of anthropogenic greenhouse gases [1]. In Europe, 90% of ammonia emissions come from agriculture, and a considerable proportion come from cattle farming [2]. Ammonia is formed when faeces and urine mix [3]. Thus, technical measures have been developed to separate faeces and urine [4], but excreta is still spread over a large area by cattle, resulting in a large emission area, which makes technical separation difficult. An innovative alternative approach would be to reduce the emission area by training cattle to use a latrine. This would also facilitate the separation of excreta and lead to improvements in animal wellbeing through better hoof and udder health, as cows would be less exposed to their excreta [5,6,7,8].

Reliable latrine use would require that cattle, like other species, learn to control a range of voluntary and reflex responses associated with toileting [9,10,11]. These include the ability to suppress impending voiding, a reflexive-like behaviour [9,11], then move to the latrine, a voluntary behaviour [12,13], and finally reinitiate voiding at the latrine. Dirksen et al. [14] reviewed the literature on the toilet training of humans and other animals and reported that cattle have the neurophysiological and cognitive capabilities for toilet training, concluding that toilet training of cattle should be possible by using operant training methods [14]. It is well known that mammals such as dogs, cats [15] and humans [11] are able to learn toileting behaviour.

For latrine (toilet) training, a sequence of new behaviours needs to be learned. An established method for training sequences is to reward a series of connected responses (behavioural chains). Chaining is a method that is often used in animal training to break down complex behavioural sequences into smaller units that can be trained step-wise using forward or backward conditioning [16,17,18]. In this process, each learned behavioural unit of the entire sequence serves as the conditioned reinforcer for the preceding behaviours and as the discriminative stimulus for the next response [19,20]. Forward chaining starts with training the first behaviour in a sequence, after which the next behavioural element is added after the first has been mastered, and so on. In backward chaining, training starts with the last behaviour of the sequence, after which the penultimate response is added, and so on [16,17,21]. The available published evidence shows no consistent differences between forward and backward chaining regarding the ease of training animals [17,18,22]. Slocum and Tiger [17] concluded that considering the high variability between studies, forward and backward chaining show similar effectiveness in teaching new behavioural chains. However, it was acknowledged that there might be differences depending on the particular task being trained. In toilet training in children, both forward and backward chaining have been used. In backward chaining, children are placed on the toilet before showing signs of voiding [23], whereas in forward chaining, voiding is interrupted by verbal instruction, after which children are moved to the toilet [24]. Toilet training can be learned via both procedures but appears to be learned faster with forward chaining [24]. Previous studies have shown that cattle are able to learn some elements of the toileting process [25,26], but the training of a full toileting sequence has not yet been achieved.

Several studies have found that learning ability is faster in younger animals than in older animals [27,28]. Other advantages of using younger animals include the ease of handling smaller animals and the possibility of utilizing the learned behaviour over their entire lifetime. For these reasons, calves were used in our study. Cattle defecate and urinate relatively infrequently (less than once per hour) [29,30,31], which makes training difficult. To train efficiently, a higher frequency of voiding is needed. It is easy to increase the frequency of urination with diuretics [25], hence urination was used for our proof-of-concept training procedure.

For practical and ethical reasons, it is not recommended to use diuretics for prolonged periods. Thus, to reduce the use of diuretics, the voluntary (non-voiding) behavioural elements of the toileting sequence were trained first. Voluntary responses were trained with a combination of backward and forward chaining (elements indicated in black in Figure 1). The voluntary elements included moving to the latrine in response to an externally applied signal, waiting in front of the reward dispenser (signalled by a flashing light), reward delivery and consumption, and exiting from the latrine. Thus, we sought to bring each element under stimulus control [32], e.g., movement to the latrine was triggered by a vibration signal and ultimately rewarded by food presentation. Subsequently, reflexive voiding responses were introduced into the chain in a manner akin to total task training [33] (elements indicated in green in Figure 1). The training of the reflexive responses involved pairing the vibration signal with the initiation of urination. An ideal outcome for the study would be initiation of movement to the latrine, firstly, in response to the externally applied stimulus (vibration) and subsequently to other behaviours occurring at the time of initiation of urination (e.g., internal stimuli associated with a full bladder). There is ample evidence in the literature from rats, pigeons and humans that initiation of behaviours can be readily transferred from one external stimulus to a novel one when the new signal occurs shortly before the first-learned stimulus (e.g., [34,35,36]). To our knowledge, transfer of control from an exteroceptive stimulus to an internal reflexive excretory signal in cattle has not previously been studied. Interestingly, evidence for the transfer of control between stimuli is provided by two different types of behavioural responses. In one case, experimental subjects orient towards the novel stimulus before the first-learned signal is presented [35] or they may orientate towards the reward location [37]. Attention to the reward location (sometimes called goal tracking) is more likely when the novel stimulus occurs in close spatial and temporal proximity to reward delivery [37,38]. When training cattle to void, Whistance et al. [26] reported that the animals orientated towards the person delivering the reward, an example of goal tracking. It was anticipated that the triggering of movement to the latrine by urination intention behaviours (and/or vibration signals) would interrupt urination and be followed by latrine entry and the reinitiation of urination (Figure 1). Thus, we expected that reflexive and operant responses could be combined to train toileting in cattle. The aim of this study was to establish a procedure for toilet training cattle using chaining methods, first by conditioning the key voluntary behaviours in the chain and then by incorporating the reflexive responses.

## 2. Materials and Methods

The study took place at the Experimental Facility for Cattle at the Leibniz Institute for Farm Animal Biology (FBN) in Dummerstorf, Germany. All procedures involving animal handling and treatment were approved by the Committee for Animal Use and Care of the Ministry of Agriculture of Mecklenburg-Western Pomerania, Germany (file reference: 7221.3-1.1-002/18).

### 2.1. Animals and Housing

Ten female German Holstein calves were purchased at an age of 30 d (range: 14–40 d) from a local commercial dairy farm, whereupon they were housed as a group in a straw-bedded pen (6.9 m × 5.7 m) and fed milk on-demand from an automatic feeder up to an age 71 d. From the 12th day after arrival at the FBN, each calf was halter trained to make later handling easier. The mean age at the start of latrine training was 91 d.

### 2.2. Experimental Facility

The training took place in a 10 × 10 m testing arena (Figure 2). The floor of the arena was covered with rubber mats. Pens and gates were constructed from steel piping (50 mm diameter). Visual contact between calves was possible except when in the designated latrine. The latrine was shielded by a solid wooden wall. The arena comprised two identical experimental training areas (one on the left, the second on the right, Figure 2). Each side contained a waiting area, a start box, an alley comprising four segments and a latrine with a bowl into which liquid rewards could be delivered (SUEVIA Mod. 20, SUEVIA HAIGES GmbH, Kirchheim/Neckar, Germany) mounted at a height of 0.6 m. Distinctive visual stimuli indicating the latrine/reward area were provided by black and yellow tape affixed to the pipework neighbouring the reward bowl. The black and yellow tape was chosen because cattle can distinguish yellow particularly well from various shades of grey [39]. The waiting area (2 m × 4 m) was connected to the start box (0.9 × 2 m), where animals were prepared prior to training (feeding, administration of diuretic). The start box was connected to the latrine (1.5 × 1 m + 2 × 2 m) via segments 3 (1.5 × 3.5 m), 2 (1.5 × 5 m) and 1 (1.5 × 2 m). Segment 4 (1.5 × 3.5 m/6 m) was also used in the training procedure. The segments were separated by manually operated gates. A remotely controlled pneumatic gate separated the latrine from segment 1 (left and right sides were operated simultaneously but could be closed individually).

The reward comprised 140 mL of a glucose-based electrolyte drink for calves referred to as Milkilyt (Milkivit, Trouw Nutrition Deutschland GmbH, Burgheim, Germany), which was mixed with water according to the manufacturer’s recommendation (50 g electrolyte and 1 L warm water) and molasses (50 mL per 1 L electrolyte mixture). The reward was continuously mixed and at a constant temperature (40 °C) in a bain-marie situated on the outside of the latrine. The reward was dispensed into the drinking bowl by a pump (which also emitted an audible signal). A blue flashing light (CO BL 70 2F, Compro® Electronic GmbH, Vechta, Germany) and an audible noise generator (RoLP, Fulleon Ltd, Cwmbran, South Wales, UK) were installed directly above the drinking bowl. Cattle can hear sounds between a range of 23–37 kHz [40]. The auditory signal used in this experiment had a frequency of 970 Hz. Furthermore, cattle have been trained to approach a feed source after an auditory signal [41]. It has been shown that cattle can distinguish yellow, pink, red, violet, blue, and green from shades of grey [39]. Thus, it was anticipated that the calves would readily react to the acoustic and visual stimuli and approach the reward bowl, which was subsequently confirmed. The reward dispensers and the visual and auditory signals on both sides of the arena were activated remotely and simultaneously by the experimenter.

From Phase 1.2 (see chapter 2.3.1.) onwards, the animals were fitted with remote-controlled vibration collars (Dogwell DW998N, TZLong Store by Amazon, Guagzhou City, China). All remote-controlled instruments were activated from a single handheld device.

The training sessions were recorded using four video cameras (AXIS M1124, Axis Communications AB, Lund, Sweden) positioned at the front and back on each side of the arena (Figure 2) and a microphone (Sennheiser MKE600; Sennheiser Electronic GmbH & Co., KG, Wedemark, Germany) positioned on the ceiling in the middle of the arena. The video and audio recordings were stored directly on a computer using Media Recorder 4.0 (Noldus Information Technology, Wageningen, The Netherlands).

### 2.3. Experimental Procedure

The calves were assigned to five training pairs matched by birth date. Within each pair, animals were assigned randomly to either the training treatment (test) or the yoked-control treatment (controls). Animals within a pair were identified by identical halter colours (blue, black, turquoise, red, and brown). Test calves were assigned randomly to either the left or right side of the training arena, and the controls were assigned to the side not occupied by the test animals. Thus, blue, turquoise and brown test calves were trained on the left and black and red on the right.

At the start of training, the calves assigned to each side were habituated as a group to the training area, and the liquid food reward for one session with a duration of one hour. During this session, the animals had access to the latrine and segments 1–3. A portion of the reward was already in the bowl, and it was refilled whenever it was fully consumed. Calves that did not drink were directed to the bowl using the halter and encouraged to drink by guiding the muzzle towards the reward. During the various phases of training, the calves had access to different areas of the testing arena (see below).

The training was divided into three main parts (training voluntary behaviours inside and outside of the latrine, and training reflexive responses). Each part consisted of two or more phases. The animals were trained once daily (called a session). During a session there were several trials, each beginning with the delivery of an exteroceptive stimulus (audible/vibration signal) and ending with reward delivery, or, in cases where there was no reaction to the signal, 10 s after the signal had been activated.

On training days (Monday to Friday), the calves were kept together in their respective waiting areas. The test and control calves were tested in a random starting order, except that the same pairs were not tested first or last on two consecutive training days. During all training phases, the control calves received all signals (e.g., vibration, sound, reward) at the same time as the test calves that were unrelated to their own behaviour. During the study, the experimenter was positioned outside of the training arena on the side occupied by the test calf to remotely activate the pneumatic gate, the various stimuli and the reward system with the handheld device. A second person was located on the other side, so the conditions were the same for both calves in a pair.

#### 2.3.1. Training Voluntary Behavioural Responses in the Latrine

The test and the control calves were confined to their respective latrine areas. Each pair of calves was trained once a day in a 30 min session, which was reduced to a 20 min session from the second week onward, as the animals’ reactions to the various stimuli decreased significantly after 20 min. The arenas were hosed down with fresh water after training each pair.

Part 1 started with magazine training (Phase 1.1) to establish an association between an audible signal and the reward (S6, Figure 1). The audible signal emanated from the reward location to draw the calf’s attention to the latrine, regardless of the direction in which it was looking. When the test calf was oriented away from the bowl, the audible signal was sounded five times in a row at a frequency of 970 Hz and a rate of 0.8 Hz (250 ms on/1 s off), followed by reward delivery. A trial was correct if the calf moved to the reward bowl after the activation of a stimulus. The next trial was initiated only after the test calf had oriented away from the bowl after reward consumption. The number of trials per training session for each pair was determined by the behaviour of the test calf. In the second training step (Phase 1.2), the audible signal was replaced by a vibration signal on a collar, as a vibration signal could be delivered without influencing the calves not currently being trained (S3, Figure 1). The transition to the vibration signal was achieved using the stimulus-fading procedure [34], in which the vibration signal was initially presented 1 s prior to the audible signal, followed by gradually increasing the delay to the audible signal, and finally presenting the vibration signal alone (Phase 1.3). After five correct trials in a row, i.e., moving towards the reward bowl within 2 s of the audible signal (Phase 1.1), after the vibration signal but before the audible signal (Phase 1.2) or within a maximum of three activations of the vibration signal (each of 1.5 s duration at intervals of approximately 1.5 s, Phase 1.3) the learning criterion for each phase was attained.

#### 2.3.2. Training Voluntary Behavioural Responses Outside of the Latrine

The goal of the next step (Phase 2.1) was to train the calves to leave the latrine after each reward delivery so that it would be possible to subsequently train latrine entry for urination. In detail, we trained the calves to leave the latrine (R1, Figure 1) after reward consumption, and to re-approach (R3, Figure 1) and re-enter the latrine (R4, Figure 1) after a further vibration signal was given (S3, Figure 1). This phase was similar to Phase 1.3, but segments 1 to 3 of the arena were also available (Figure 2). The vibration signal was given when the calf stopped walking in segments 1 to 3, ideally with its body oriented in the direction of the latrine to facilitate immediate movement towards it. The learning criterion was five consecutive correct responses (approach towards and entry into the latrine) to the activation of the vibration signal. Test calves that did not meet this criterion were moved to the next phase after eight sessions.

In Part 3 (training of urination), the calves were expected to learn to wait in the latrine until the completion of urination before receiving any reward. The duration of urination in cattle is usually less than 30 s [42,43,44]. Thus, the purpose of this next step (Phase 2.2) was to prepare the calves for a delay between latrine entry and reward delivery. To provide a signal that a reward was imminent, a blue flashing light above the dispenser (S5, Figure 1) was activated for the duration of the delay. The delay duration was increased stepwise from 2 s to 30 s (latencies of 2, 4, 7, 11, 15, 20, 25, and 30 over successive days). The calf was required to stand calmly in front of the bowl during the delay; otherwise, the reward was not delivered. All four segments and the latrine were available to the calves in this Phase.

#### 2.3.3. Incorporation of Reflexive Responses into the Chain

The goal of Part 3 was to introduce reflexive responses into the previously trained behavioural chain first in the latrine area only (Phase 3.1) and then in the alley connected to the latrine (Phase 3.2).

Five minutes before the start of a training session, while standing in the start box, the test and control calves were administered 1.3 mL (1.5 mL in the second week) of diuretic (Diuren, Wirtschaftsgenossenschaft deutscher Tierärzte (WDT), Garbsen, Germany), into the jugular vein to increase urination frequency. The training in Part 3 covered periods of two to three days of training with one session per day, interspersed with two to three days without training to prevent undue injury to the jugular vein. In contrast to the previous phases, the number of training sessions in Part 3 was predetermined, as the total number of days on which diuretics could be administered was limited to ten by the Committee for Animal Use and Care. Thus, Phase 3.1 comprised three sessions, and Phase 3.2 comprised seven sessions.

In Phase 3.1, the calves were locked in the latrine, and the vibration signal was activated at the first sign of urination, e.g., raising the tail, spreading the hind legs and/or arching the back. A reward was delivered if i) the calf interrupted urination, approached the reward bowl, and reinitiated urination while waiting in front of the bowl or ii) the calf oriented towards the reward bowl during urination. Otherwise, no reward was delivered.

Waiting in front of the bowl was accompanied by the blue flashing light (waiting signal). It became clear that the reinitiation of urination did not occur regularly within 30 s after approaching the reward bowl. Therefore, this phase was discontinued after three sessions, and the flashing light (see above) was not used in Phase 3.2.

In Phase 3.2, the calves had access to the latrine and segment 1 for the first six training sessions and access to segments 2 and 3 in the final (7th) session. Starting with an ‘intention to urinate’ response by a calf (either in the latrine or the alley), six variants of a behavioural chain were possible. Each variant indicated a different level of training success (as shown in Figure 3). If the calf was in the alley (segment 1–3) when it showed the intention to urinate, then sequences (Seq) 1–5 were possible. Situations in which a calf was already in the latrine at the beginning of a urination event were denoted by Seq6+. In all sequences, when calves entered the latrine after receiving a vibration signal, they were confined for up to 5 min (by closure of the pneumatic gate) to increase the probability of urination in the latrine. The control calves were confined in the latrine simultaneously or as soon as they were in the latrine. The behaviour of the control calves was also assigned to one of the six sequences if the appropriate criteria were met.

Seq1+ and Seq3+ were deemed ‘correct’ responses, as urination in the alley was totally withheld (Seq1+) or interrupted (Seq3+), followed by walking to the latrine, the reinitiation of urination in the latrine and the delivery of a reward.

Seq2-, Seq4-, and Seq5- were not rewarded. In Seq2-, urination was withheld in response to the vibration signal. In Seq4-, urination was stopped either because of interruption or because the bladder was emptied. Both sequences included movement into the latrine but not the reinitiation of urination and were thus designated as an ‘incomplete’ response. In Seq5-, the calves did not inhibit or interrupt urination (in response to the vibration signal) or walk to the latrine (thus being designated a ‘false’ response).

In Seq6+, the calves were already situated in the latrine when urination intention behaviours occurred. Thus, the tactile signal was not activated, and completed urination events were rewarded. For Seq6+, however, it was not clear if the animal had previously entered the latrine in anticipation of urination or if it was situated in the latrine for other reasons (e.g., to be in close proximity to the reward site). Thus, it was unclear whether the responses in Seq6+ demonstrated control of urination or not (thus, being designated an ‘ambiguous’ response).

### 2.4. Data Analysis

All videos were analysed by a single person using The Observer® XT13 software (Noldus Information Technology, Wageningen, The Netherlands). Due to technical problems, there were no videos available for the black pair for session 1 of Phase 3.1 and session 5 of Phase 3.2.

The following events were coded for the test and control calves: (a) start of the trial with the trial number and location of the calf, (b) end of the trial with the trial number and the result (correct/incorrect), (c) reward delivery (yes/no), (d) duration of the flashing light, (e) confinement in latrine. Events (a) and (b) were coded for Parts 1 and 2 only, and event (e) was coded for Part 3 only. The reason for this was that in Part 3, no trials were defined that started with an exteroceptive stimulus, instead trials started with the intention to urinate. Similarly, events (a) and (b) were not coded in Part 3. To take into account the time and duration of confinement in the latrine, event (e) was coded in Part 3 only. The behaviours recorded in Part 3 are shown in an ethogram (Table 1). Intraobserver reliability was examined by re-coding 25% of the videos from Phase 3.2 by the original observer [45,46] because this was the phase including the complete behavioural sequences. Cohen’s kappa was calculated with the routine included in The Observer® XT13 software with *κ* = 0.91 for the frequency, with regard to the sequence of events over all observed behaviours. Cohen’s kappa for duration, with regard to the sequence of events over all observed behaviours, was also *κ* = 0.91 [47]. Cohen’s kappa values between 0.81 and 1.00 are regarded as almost perfect [48].

Due to the small sample size, some of the data are presented descriptively. The calculation of mean values was performed first for each calf and then across all calves in each session or phase.

Differences in the number of correct trials during voluntary behaviour training in Part 1 between the test and control calves were analysed using the Kruskal-Wallis test in Jamovi (Version 1.1.9.0, https://www.jamovi.org) which is based on R. The duration of urination by the test calves in the different behavioural sequences within or outside the latrine was analysed using the Friedman test in Jamovi (Version 1.1.9.0, https://www.jamovi.org). Differences in the duration (total and individual events) and frequency of urination between the test and control calves were analysed using one-way ANOVA according to the MIXED procedure in SAS (Version 9.4, SAS Institute Inc., Cary, NC). In the MIXED procedure, the calf was included as a repeated factor, and the treatment (test/control) was the fixed effect.

## 3. Results

### 3.1. Training Voluntary Behavioural Responses in the Latrine

The number of trials required to meet the criterion for the test calves in Part 1 is shown in Table 2. All calves met the learning criteria, but there were differences between individuals in the number of trials required to meet the criteria. In Phase 1.1, for all test calves, the association between the audible signal and the reward delivery was learned in 11 ± 5.2 trials (mean ± SD). The calves learned to respond consistently to the combined vibration and auditory signals after an average of 14.6 ± 10.9 trials (Phase 1.2). Finally, the association between the vibration signal alone and reward delivery (Phase 1.3) was learned in an average of 8.8 ± 3.4 trials. The number of correct trials among the test and the control calves are shown in Table 3. The numbers were significantly higher in the test calves compared to the controls (Phase 1.1: *χ*^2^ = 7.03, *df* = 1, *p* = 0.008; Phase 1.2: *χ*^2^ = 4.53, *df* = 1, *p* = 0.033; Phase 1.3: *χ*^2^ = 6.40, *df* = 1, *p* = 0.011).

### 3.2. Training Voluntary Behavioural Responses Outside of the Latrine

Four of the five test calves learned to leave the latrine and wait in the alley prior to the delivery of a subsequent vibration signal (Phase 2.1) (mean ± SD: 11.25 ± 7.1 trials). After eight sessions (39 trials), the turquoise calf had not met the formal learning criterion (but progressed to the next phase as specified in the training schedule). Figure 4 shows that, in Phase 2.1, the test calves were outside the latrine at the start of 98% of all trials, while the control calves were in the latrine or with the muzzle in the bowl at the start of 75% of all trials.

In Phase 2.2, four (blue, black, turquoise, and brown) out of five calves learned to wait up to 30 s for the reward (with ≥80% correct trials) while the flashing light was activated. The remaining calf (red) learned to wait for up to 15 s, with ≥80% correct trials.

### 3.3. Incorporation of Reflexive Responses into the Chain

Table 4 shows the number of urination events in which the calves oriented towards the reward bowl and the total number of urination events in Phase 3.1. In this phase, by the third session, four (Blue, Black, Turquoise, Brown) of the five test calves oriented towards the bowl during urination on 100% of occasions (Table 4). The Red test calf oriented towards the reward bowl on 50% of occasions by the third session. After stopping urination, the test calves did not reinitiate urination in front of the bowl within 30 s. On one occasion, a control calf (Blue) urinated at the same time as a test calf, so it also received the vibration signal during urination.

#### 3.3.1. Behavioural Sequences

In the seven sessions in Phase 3.2, we observed 94 urinations by the test calves and assigned each to one of the six different sequences (Table 5 and Table 6). Seventy-five percent of all sequences terminated with urination in the latrine (Seq1+, Seq3+, Seq6+) (Table 6). Forty-four percent were ‘correct’ (Seq1+ and Seq3+), 22% were ‘incomplete’, (Seq4-), 31% were ‘ambiguous’ (Seq6+) and 3% were ’false’ (Seq5-) (Table 6). The most common sequences were Seq3+ and Seq6+, which were observed in all calves except for Seq6+ in Black. Seq4- was also observed in all calves. Seq2- never occurred.

Sixty-five sequences were initiated outside the latrine (i.e., all sequences except for Seq6+); 62 (95%) of these resulted in latrine entry (Seq1+, Seq3+, and Seq4-), and 41 (65%) were followed by urination in the latrine (Seq1+ and Seq3+). The three events that did not result in latrine entry (Seq5-) involved only one calf (turquoise) and occurred early in training in Phase 3.2.

There appeared to be trends in the relative frequency of the different sequences across the sessions. These trends were observed in the comparison of the relative occurrence in the first and last two sessions (from Table 5). Seq3+ declined from 43% of sequences to 11%. Seq4- increased from 7% of sequences to 32%. Seq6+ increased from 27% of sequences to 42%. For the remaining sequences, there were no apparent trends across sessions.

In Phase 3.2, 63 urination events were observed in the control calves. Among these events, 42 occurred in the latrine and 18 in the alley, and during three events, the calves entered the latrine while urinating. Since the control calves rarely urinated simultaneously with the test calves (six events), most events did not match with any of the six sequences. There were three Seq6+ sequences and one Seq4- sequence in the controls. For the remaining two events, the vibration signal was given when the control calf was urinating in the latrine or the reward was delivered just at the beginning of the urination event in the latrine.

#### 3.3.2. Duration of Urination and Time to Reinitiate Urination

For the intention to urinate, a duration of 2 ± 1 s was recorded over all test calves in all sequences. For urination in the alley, the duration for the test calves was 5 ± 2 s (mean ± SD) in Seq3+ compared to 10 ± 1 s in Seq4- (*χ*^2^ = 5, *df* = 1, *p* < 0.025). In Seq5-, where alley urination was not inhibited or interrupted, the mean duration of urination was 14 ± 1 s.

For urination in the latrine, the mean duration was 13 ± 5 s in Seq1+, 9 ± 2 s in Seq3+ and 10 ± 1 s in the case of Seq6+. The duration of urination in the latrine did not differ between the sequences (*χ*^2^ = 1, *df* = 2, *p* = 0.607).

The time from latrine entry to the (re-)initiation of urination was 77 ± 31 s in Seq1+, 164 ± 78 s in Seq3+, and 99 ± 69 s in Seq6+ (*χ*^2^ = 3, *df* = 2, *p* = 0.223).

The frequency of all urination events in a session was higher in the test calves (4.1 ± 0.5) (LSM ± SE) than in the control calves (1.9 ± 0.5) (F_1,8_ = 11.36, *p* < 0.01). The mean duration across all urination events for the test calves was 9 ± 1 s, which was approximately half of the duration for the control calves (17 ± 1 s; F_1,8_ = 46.95, *p* < 0.001). There was no difference in the sum of the duration of all urination events for an animal in one session between the test (35 ± 5 s) and control calves (32 ± 5 s) (F_1,8_ = 0.13, *p* = 0.726).

#### 3.3.3. Association of Urination (and Place) with Reward

For sequences in which urination occurred in the latrine (Seq1+, Seq3+, Seq6+), the calves displayed goal tracking (orienting to the reward dispenser) on 63 ± 14% (mean ± SD) of urinations. This orientation behaviour was shown by all calves.

#### 3.3.4. Location of the Calves

The test calves spent 60 ± 7% of their time in the latrine, while the control calves spent 72 ± 7% of their time there. However, time spent in the latrine was determined by different factors in the test and control calves (see Materials and Methods).

## 4. Discussion

The aim of this study was to establish and evaluate a procedure for toilet training cattle using chaining methods, first by conditioning the key voluntary (operant) behaviours in the chain, then by adding reflexive responses using a variant of a procedure that has been used successfully to train toileting in children. A novel finding of this study was that test calves successfully learned the voluntary elements of the behavioural chain, and reflexive responses were also incorporated into the chain. This was demonstrated by the 95% of urination events in which the test animals inhibited urination or otherwise responded to the tactile signal (while positioned outside the latrine) and subsequently moved towards and entered the latrine. Furthermore, in 65% of the sequences that started in the alley, urination was reinitiated in the latrine. The absence of learning in the control calves provides additional support for non-accidental learning by the test calves. The failure of the controls to learn is not surprising, as the stimuli and food delivery in the controls were not reliably associated with the appropriate voluntary and reflexive behaviours.

### 4.1. Initiation and Completion of Voluntary Behaviour Chains

Our results extend previous findings that cattle can be conditioned using operant methodologies [12,49] to the learning of long chains of behaviour. All test calves learned the full behaviour chain, including leaving the latrine, stopping in the alley and waiting for a vibration signal before re-entering the latrine, then waiting in front of the bowl before receiving a reward. Two different types of external stimuli (auditory and vibration) were used to initiate the operant chains, beginning in the alley. Both were effective chain initiators in test calves but not control calves, demonstrating that (i) stimulus control was readily achieved with audible and tactile stimuli in cattle and that (ii) the transfer of control from one type of stimulus (auditory) to another occurred quickly. Evidence of auditory stimulus control was provided by the test calves reliably approaching the reward dispenser in response to sound presentation. The stimulus control was transferred readily from the auditory to a tactile stimulus, as the calves moved to the reward bowl in response to the vibration signal and before the sound was delivered. Stimulus control has been demonstrated in cattle in other studies with both auditory and tactile stimuli [41,50]. For example, Wredle et al. [41] trained heifers to approach a feed source after an auditory signal. In another study with cattle, a vibration signal was used [50] to signal approach to a feeding station. It is not surprising that the external stimuli failed to initiate the operant behaviour chain in control calves, as unlike the test animals, the signals were delivered independent of a particular behaviour or location. In most instances, the controls were located in the latrine at the time of signal activation and, thus, did not display the correct/rewarded behavioural sequences. This is not surprising as the controls were not explicitly trained to leave the latrine.

Our study raises the possibility that the initiation of movement to the latrine can also be triggered by internal pre-voiding signals, as has been reported with toilet training in children [24]. This notion is supported by (i) the decline over sessions in the relative frequency of sequences in which urination in the alley was interrupted by the external signal (Seq3+) and a concomitant increase in the frequency of sequences in which the calves entered the latrine shortly before urinating (Seq6+) without a prior signal (although see later for a caveat) and (ii) the very low incidence (3%) of sequences showing an absence of learning (Seq5-). Furthermore, internal (pre-)voiding signals influenced the probability of orientation towards the reward location in the latrine, providing additional evidence that these signals can serve as discriminative stimuli and guide voluntary responses in cattle. These results build on the findings of Whistance et al. [26], who reported that adult cows rewarded for voiding looked or moved towards the source of reward delivery (a person) during voiding. However, the cattle in the Whistance et al. [26] study did not move to the designated latrine area for voiding, most likely because the animals were not rewarded outside of the training period for excreting in the latrine, preventing the transfer of stimulus control to the correct voiding location. Further studies are needed to confirm that latrine seeking/voiding in cattle can be brought under the control of pre-voiding signals. Such evidence would increase the likelihood that cattle could be readily toilet trained outside of laboratory settings without the need for delivering external signals, e.g., vibration.

During correct toileting sequences, in the current study, the performance of excretory responses at the latrine introduces a delay between latrine entry and reward delivery. Reinforcement efficacy declines exponentially with increases in reward delay [51], so it is important that delays due to voiding do not compromise learning. Here, we show for the first time that delay intervals that were approximately twice as long as that required for urination did not impair operant learning in calves. For the calves in our study, the possibility that they might be able to wait for a reward even longer than 30 s cannot be excluded. Our aim was not to test impulse control in calves, but these results indicate that calves have a degree of self-control, which would be interesting to explore further.

### 4.2. Control and Learning of Reflexive Responses

The control of voiding by inhibiting (or interrupting) in the wrong place and reinitiating excretory responses in the right place (latrine) is a key element in the successful learning of toilet use in other species, e.g. humans and dogs [14]. An original finding from our study is that calves can inhibit urination in response to an external stimulus delivered just prior to voiding. There was variation between the test calves in the degree of the inhibition of urination in the alley; two of the five animals did not inhibit urination (i.e., did not show Seq1+). The absence of inhibition may have been due to a failure to learn to control reflexive voiding responses, but this is unlikely, as all calves regularly interrupted urination in response to the external stimulus on other occasions (Seq3+), then entered the latrine and reinitiated urination. A more likely explanation for the lack of inhibition in some calves was that the stimulus was delivered too slowly during the urination intention responses (which had a duration of 2 s or less). The ability of the calves to inhibit urination, together with the demonstration that calves can interrupt urination, suggests control over the sphincter muscles [9]. The possibility that cattle might be able to interrupt voiding was also observed by Whistance et al. [26]. They reported that heifers interrupted voiding in a small percentage (8%) of behavioural sequences when moving towards the trainer (and source of reward).

In children, one goal of toilet training is to transfer the control of voiding inhibition and toilet entry from the instructor to the child, with toilet visits initiated by impending (internal) excretory stimuli, followed soon thereafter by voiding in the toilet [24] (similar to Seq6+ in our study). In 31% of all sequences in our study, the animals were already in the latrine at the time that they showed an intention to urinate (Seq6+). There are three possible reasons for this. The calves may have (i) entered the latrine in response to internal stimuli signalling impending voiding; (ii) chosen to maintain close proximity to the site of reward delivery; and (iii) been situated in the latrine by chance. Evidence of learning is provided by the short time from latrine entry to urination (99 s), which was comparable to that shown by calves that inhibited voiding when externally signalled and reinitiated voiding in the latrine (77 s, Seq1+). The Whistance et al. [26] study provides some additional support for the idea that cattle can respond appropriately to internal pre-voiding cues—in 9% of sequences the heifers moved to the trainer before voiding. The higher rate of responding to reflexive cues in our study (31%) is likely attributable to differences in the training regimes e.g. all correct responses in the latrine were rewarded in our study, whereas this would not have been the case in the Whistance et al. [26] study. Evidence of the effects of reward location and/or chance causing Seq6+ is provided by the high proportion of time spent in the latrine. It is not possible to disentangle these possibilities, but the evidence suggests that all three factors likely contributed to latrine voiding in Seq6+.

In addition, the procedure used in our study has features in common with those used by Van Wagenen et al. [24] in children. In the Van Wagenen et al. [24] study, disabled children were startled with a verbal stimulus to induce them to stop urination and then taken to the toilet to reinitiate urination. The startle likely inhibited urination via involuntary reflex pathways [14]. In contrast, the stimulus used in our study was a benign signal, which followed training with positive reinforcement, initiating voluntary movement towards the latrine. This indicates that the activation of the sphincter muscles to inhibit urination in calves does not require triggering by an involuntary alarm response. Rather, it seems that voiding reflexes in cattle can be controlled voluntarily. This is supported by the comparison of the duration of urination between the test and control calves. The total duration of urination in a session was similar for the test and control calves, but each urination event was shorter and the frequency greater for the test calves.

Our study provides some evidence that the reinitiation of urination in cattle is under voluntary control. The primary evidence arises from the observation that all sequences beginning with signalled inhibition (Seq1+) in the alley terminated with urination in the latrine a short time after latrine entry. This is supported by evidence from humans showing that if voiding can be inhibited or interrupted, then it can be reinitiated [11]. The time to reinitiate urination in the latrine was longer after interruptions (Seq3+) than after inhibitions in the alley (Seq1+). However, we cannot be sure whether reinitiation after interruption is under voluntary control or is controlled reflexively in response to refilling of the bladder [11]. Vaughan et al. [25] showed that urination in calves can also be brought under the (stimulus) control of a place (latrine) using positive reinforcement, providing additional support for the idea that reflexive behaviours can be conditioned using operant procedures [14].

There was individual variation between calves regarding the probability of reinitiating urination in the latrine. One possible explanation is a failure to learn the voiding location. However, this is unlikely, as there was a high proportion of correct responses among all calves (where urination was reinitiated in the latrine). A more plausible explanation is that the bladder contained insufficient urine for voiding to occur in the latrine following partial emptying in the alley, since the duration of pre-entry urination was longer when voiding was not reinitiated in the latrine (Seq4- compared to Seq3+). Furthermore, as mentioned previously, all calves frequently oriented towards the reward bowl during voiding in the latrine, demonstrating stimulus control of voiding in this location (a learned process [32]). Partial extinction of the latrine entry response may also explain the absence of consistent reinitiation of urination in the latrine, as there was a considerable delay from entry to reward delivery. The delay varied between 77 s (Seq1+, inhibited voiding sequences) and 164 s (Seq3+, interrupted voiding sequences) [51].

For cattle, toilet training might be facilitated by a modification of the procedure in which the reflexive behaviour is first trained to occur in the correct location (latrine) before adding the voluntary behavioural elements, including approach to the latrine. This alternative method potentially increases the strength of the learned association between voiding and toileting location. It has also been used successfully with children [23]. Other possibilities for facilitating training include making the latrine more clearly distinguishable from the alley (e.g., by using a unique colour scheme for the latrine walls and floor) and methods for reducing the time spent in the latrine.

## 5. Conclusions

Our training procedure was a variant of a common approach used for toilet training in children, in which excretory behaviour is interrupted, and its resumption is rewarded [24]. These results showed that calves can be toilet trained by incorporating reflexive voiding (urination) responses into a learned voluntary behavioural chain. In farming practice, cattle use of latrines would help in reducing harmful gaseous emissions [4] and improving the hygiene of living quarters [6,8].

## Figures and Tables

**Figure 1 animals-10-01889-f001:**
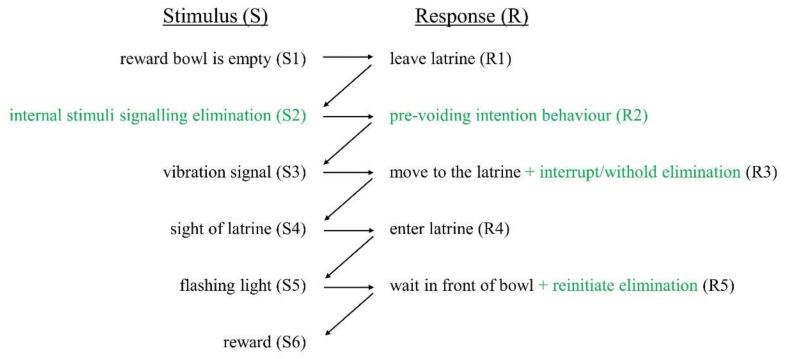
Behavioural chain for training latrine use in cattle. First, the voluntary behaviours (indicated in black) are to be trained, after which the voiding-related behaviours (indicated in green) are added to the chain.

**Figure 2 animals-10-01889-f002:**
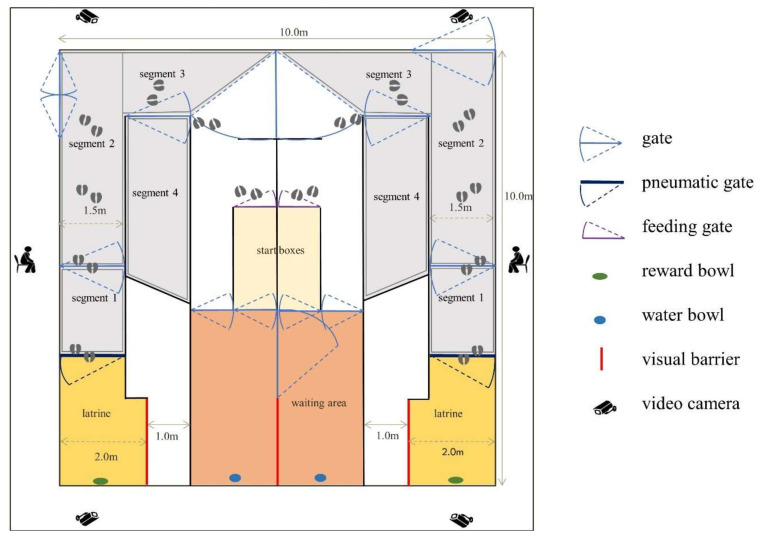
Testing arena in which the calves were trained. Each side was identical and contained a waiting area (shown in red) and a start box with a self-locking feeding gate, where the calves were prepared for training. Training occurred in the latrine (shown in yellow) and in segments 1 to 4 (shown in grey). Cameras placed at either end of the arena on each side were used to record the training sessions.

**Figure 3 animals-10-01889-f003:**
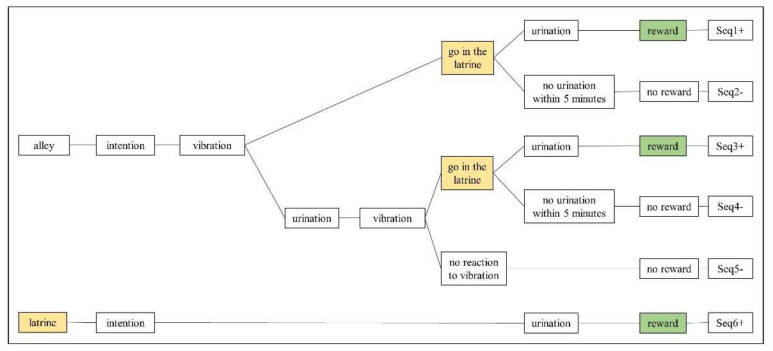
Variants of behavioural chains in Phase 3.2. Sequences that started in the alley and ended with urination in the latrine (Seq1+, Seq3+) were rewarded and designated as ‘correct’. Other sequences that started in the alley were designated as ‘incomplete’ if the calf moved to the latrine but did not reinitiate urination within 5 min (Seq2-, Seq4-) or were designated as ‘false’ if the calf remained in the alley (Seq5-). If the calf was already in the latrine when urination occurred, the reward was delivered (Seq6+), and this sequence was designated as ‘ambiguous’. For consistency, the colour of the latrine (yellow) and reward text boxes (green) were selected to match the related areas in Figure 2.

**Figure 4 animals-10-01889-f004:**
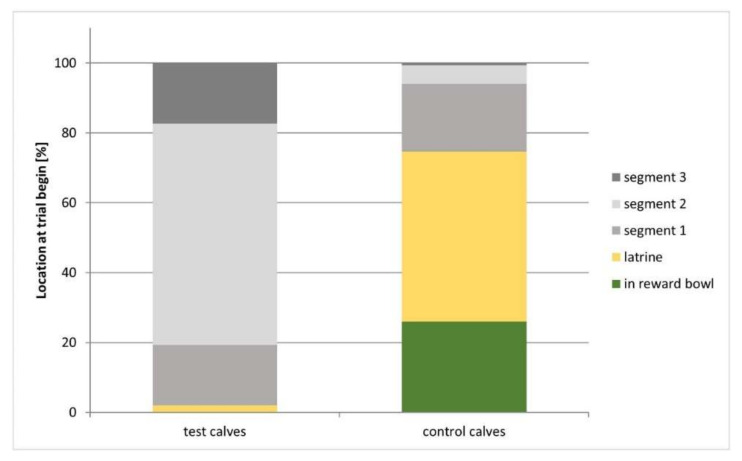
Location of the test calves and the controls at the start of the trials in Phase 2.1. In this phase, learning would be demonstrated by the calf exiting the latrine, moving along the alley and waiting before delivery of a vibration signal.

**Table 1 animals-10-01889-t001:** Ethogram for the video analysis of behaviours during reflexive behaviour training in Part 3.

Recording Type ^1^	Behavioural Parameter	Definition
D, F, T	intention (to urinate)	arching the back, raising the tail and/or spreading the hind legs
D, F, T	urination	urinating (urine is voided)
D, F, T	location of the calf	>50% of the calf’s body is in a given segment or in the latrine
T	orientation towards the bowl	moving the head in the direction of the bowl, and/or starting locomotion towards the bowl (only during a urination event)

^1^ recording types: duration (D) in s, frequency (F), and time of occurrence (T) in relation to the start of the video in s.

**Table 2 animals-10-01889-t002:** Individual number of trials until meeting the criteria in Part 1 for the test calves.

Phase	Blue	Black	Turquoise	Red	Brown
Phase 1.1	6	18	14	11	6
Phase 1.2	5	19	13	5	31
Phase 1.3	8	7	10	14	5

**Table 3 animals-10-01889-t003:** Individual number of correct trials of the test and control calves in Part 1.

Phase	Blue	Black	Turquoise	Red	Brown
	Test/Control	Test/Control	Test/Control	Test/Control	Test/Control
Phase 1.1	5/2	12/0	10/3	5/2	5/1
Phase 1.2	5/0	12/8	8/0	5/1	20/0
Phase 1.3	6/4	7/2	7/0	13/5	5/0

**Table 4 animals-10-01889-t004:** Number of urination events with orientation towards the bowl and the total number of urination events in Phase 3.1 during the sessions for the test calves.

Test Calf	Session 1		Session 2		Session 3	
	Events with Orientation	Total Number of Events	Events with Orientation	Total Number of Events	Events with Orientation	Total Number of Events
Blue	0	3	2	3	8	8
Black	1	2	2	3	4	4
Turquoise	3	5	4	4	9	9
Red	1	3	0	3	2	4
Brown	3	3	9	9	6	6

**Table 5 animals-10-01889-t005:** Total number of occurrences of each sequence in each session (1 to 7) in Phase 3.2 for the test calves.

Session	Seq1+	Seq2-	Seq3+	Seq4-	Seq5-	Seq6+
Session 1	2	0	6	2	2	4
Session 2	3	0	7	0	0	4
Session 3	3	0	5	5	1	6
Session 4	1	0	5	5	0	1
Session 5	0	0	4	3	0	6
Session 6	1	0	1	4	0	3
Session 7	2	0	1	2	0	5

**Table 6 animals-10-01889-t006:** Total number of occurrences of each sequence for each test calf across all sessions (1 to 7) in Phase 3.2.

Test Calf	Seq1+	Seq2-	Seq3+	Seq4-	Seq5-	Seq6+	Total
Blue	0	0	6	6	0	6	18
Black	6	0	2	1	0	0	9
Turquoise	4	0	11	6	3	1	25
Red	0	0	3	4	0	14	21
Brown	2	0	7	4	0	8	21
total	12(13%)	0(0%)	29(31%)	21(22%)	3(3%)	29(31%)	94(100%)

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
