# Peer review of "How Can Cattle Be Toilet Trained? Incorporating Reflexive Behaviours into a Behavioural Chain"

_animals, 2020, doi:10.3390/ani10101889_

Round 1
Reviewer 1 Report
This is an interesting study essentially trying to potty train baby cows. Previous work suggests cows are capable of this type of training. I think this is an interesting goal in itself. The authors begin by arguing that this could basically help reduce methane emissions, but this is a bit of a stretch. It is not only unclear how exactly this would work, but how practical this really is. The authors might consider starting the paper differently. Overall, I found this to be a clear and well-written paper. I do not have many substantive concerns, except for some issues with the data analysis. First, there is no such thing as a non-parametric ANOVA. What test did you actually use? Similarly, there is no way that I know of to do repeated measures in a non-parametric test like this. You need to provide more details on your statistics, including the actual test statistics and samples sizes/degrees of freedom in the Results. Right now, you only do this for the ANOVA. You need to give test statistics and sample sizes for the non-parametric stats, which I assume are Kruskal-Wallis tests (which cannot have repeated measures). Currently, this is all a bit opaque and it needs to be more transparent. Once this is sorted, I think the manuscript will be ready for publication.
Author Response
Dear reviewer
In the point-to-point reply, we first specify the line number in the original manuscript to which the comment refers, as well as the comment itself.
The answer is indicated in red. We indicate the line number in the revised manuscript and the changes we have made at this point in the revised manuscript.
Reviewer 1:
L 50: The authors begin by arguing that this could basically help reduce methane emissions, but this is a bit of a stretch.
Response: We agree that the second sentence of the introduction is misleading. (L 49) To rectify, we have deleted the sentence.
L 319-320: First, there is no such thing as a non-parametric ANOVA. What test did you actually use?
Response: You are right. We have used the Kruskal-Wallis test here, which is also called non-parametric ANOVA in various program packages for statistical testing of data (https://en.wikipedia.org/wiki/Kruskal%E2%80%93Wallis_one-way_analysis_of_variance; https://www.jamovi.org/jmv/anovanp.html).
(L 331) For better understanding we have changed the name of the test to Kruskal-Wallis test.
L 321-322: Similarly, there is no way that I know of to do repeated measures in a non-parametric test like this.
Response: The Friedman test is a non-parametric statistical test often called as repeated measures ANOVA (https://en.wikipedia.org/wiki/Friedman_test; https://www.jamovi.org/jmv/anovarmnp.html).
(L 333) For better understanding we have changed the name of the test to Friedman test.
L 319-326: You need to provide more details on your statistics, including the actual test statistics and samples sizes/degrees of freedom in the Results. Right now, you only do this for the ANOVA. You need to give test statistics and sample sizes for the non-parametric stats, which I assume are Kruskal-Wallis tests (which cannot have repeated measures). Currently, this is all a bit opaque and it needs to be more transparent.
Response: (L 348-349) The full test statistics for the Kruskal-Wallis test have been added. (L 407, L 411, L 413) This was also done for the Friedman test

Reviewer 2 Report
Line 97: Error message instead of a reference
Lines 104-114: Difficult to understand this when one is not familiar with terminology of animal behaviour and training- consider breaking into a few smaller sentences, and defining a few terms?
Lines 209-215: Defining a "Trial" could either come at the very start of this section, before you start describing phases, or may be better in the next section, after Line 303, when you start talking about coding events for each trial.
Line 308: Why were certain events recorded for only certain phases? An extra line to explain this may help.
Line 330: Does "and trials" need to be here or is it a typo error?
Overall this was a fascinating read. I struggled with understanding the exact sequence in the Methods section, possibly due to my unfamiliarity with behavioural work. I'd like to suggest a few extra lines at the start of the Methods section, giving a broad overall structure of the Method: defining what is a 'session', a 'trial', a 'part' and a 'phase', and what each step will achieve- maybe around line 192. Other than that, this was a succinct and scientifically sound piece of work and I hope it will add value to this area of research.
Author Response
Dear reviewer
In the point-to-point reply, we first specify the line number in the original manuscript to which the comment refers, as well as the comment itself.
The answer is indicated in red. We indicate the line number in the revised manuscript and the changes we have made at this point in the revised manuscript.
L 97: Error message instead of a reference
Response: (L 95) We have corrected the error.
L 104-114: Difficult to understand this when one is not familiar with terminology of animal behaviour and training- consider breaking into a few smaller sentences, and defining a few terms?
Response: Modified as below:
(L 102-112) “An ideal outcome for the study would be initiation of movement to the latrine, firstly, in response to the externally-applied stimulus (vibration) and subsequently to other behaviours occurring at the time of the initiation of urination (e.g. internal stimuli associated with a full bladder). There is ample evidence in the literature from rats, pigeons and humans that initiation of behaviours can be readily transferred from one external stimulus to a novel one when the new signal occurs shortly before the first-learned stimulus (e.g., [34-36]). To our knowledge, transfer of control from an exteroceptive stimulus to an internal reflexive excretory signal in cattle has not previously been studied. Interestingly, evidence for the transfer of control between stimuli is provided by two different types of behavioural responses. In one case, experimental subjects orient towards the novel stimulus before the first-learned signal is presented [35], or they may orientate towards the reward location [37].”
L 192: I'd like to suggest a few extra lines at the start of the Methods section, giving a broad overall structure of the Method: defining what is a 'session', a 'trial', a 'part' and a 'phase', and what each step will achieve- maybe around line 192.
Response: We have amended the Methods as follows in line 197-202: “The training was divided into three main Parts (training voluntary behaviours inside and outside of the latrine, and training reflexive responses). Each Part consisted of two or more Phases. The animals were trained once daily (called a session). During a session there were several trials, each beginning with the delivery of an exteroceptive stimulus (audible/vibration signal) and ending with reward delivery, or, in cases where there was no reaction to the signal, 10 s after the signal had been activated.” The information about achievements is already included in the text in L 216-217, L 224-226, L 234-235, L 244-245, L 254-255.
L 209-215: Defining a "Trial" could either come at the very start of this section, before you start describing phases, or may be better in the next section, after Line 303, when you start talking about coding events for each trial.
Response: (L 199-202) The definition of a trial is now presented earlier (at the beginning to the experimental procedures section).
Line 308: Why were certain events recorded for only certain phases? An extra line to explain this may help.
Response: We added the following sentences: (L 317-320) “The reason for this was that in Part 3, no trials were defined that started with an exteroceptive stimulus, instead trials started with the intention to urinate. Similarly, events a) and b) were not coded in Part 3. To take into account the time and duration of confinement in the latrine, event e) was coded in Part 3 only.”
Line 330: Does "and trials" need to be here or is it a typo error?
Response: (L 341) We have deleted this.

Reviewer 3 Report
Dear authors,
This is an interesting and very relevant study and I acknowledge your efforts of endeavouring into this topic! Although I understand this topic being tricky to explain, the article is generally long and the text flow could benefit from more consistent wording, and shorter/simpler sentences to assist the reader. In order to do so, please consider to revise the text to be clear to all readers, also those novel to the cognitive mechanisms and tests used. Please also consider to include and explain the underlying reasons for why you chose to use the external stimuli/cues that is used (blue light, sound and vibration). As your sample size is low (and as you also find individual variation), I also urge you to clearly state the limitations of your results, e.g. by emphasising and discussing these more in your discussion. Finally, please refine your conclusions accordingly.
Please see the detailed comments in the attached pdf document, which I hope may assist you in the revisions.

Author Response
Dear reviewer
In the point-to-point reply, we first specify the line number in the original manuscript to which the comment refers, as well as the comment itself.
The answer is indicated in red. We indicate the line number in the revised manuscript and the changes we have made at this point in the revised manuscript.
Reviewer 1:
L 50: The authors begin by arguing that this could basically help reduce methane emissions, but this is a bit of a stretch.
Response: We agree that the second sentence of the introduction is misleading. (L 49) To rectify, we have deleted the sentence.
L 319-320: First, there is no such thing as a non-parametric ANOVA. What test did you actually use?
Response: You are right. We have used the Kruskal-Wallis test here, which is also called non-parametric ANOVA in various program packages for statistical testing of data (https://en.wikipedia.org/wiki/Kruskal%E2%80%93Wallis_one-way_analysis_of_variance; https://www.jamovi.org/jmv/anovanp.html).
(L 331) For better understanding we have changed the name of the test to Kruskal-Wallis test.
L 321-322: Similarly, there is no way that I know of to do repeated measures in a non-parametric test like this.
Response: The Friedman test is a non-parametric statistical test often called as repeated measures ANOVA (https://en.wikipedia.org/wiki/Friedman_test; https://www.jamovi.org/jmv/anovarmnp.html).
(L 333) For better understanding we have changed the name of the test to Friedman test.
L 319-326: You need to provide more details on your statistics, including the actual test statistics and samples sizes/degrees of freedom in the Results. Right now, you only do this for the ANOVA. You need to give test statistics and sample sizes for the non-parametric stats, which I assume are Kruskal-Wallis tests (which cannot have repeated measures). Currently, this is all a bit opaque and it needs to be more transparent.
Response: (L 348-349) The full test statistics for the Kruskal-Wallis test have been added. (L 407, L 411, L 413) This was also done for the Friedman test.
Dear reviewer
In the point-to-point reply, we first give the line number in the original manuscript to which the comment refers and the comment itself.
Then we give the line number in the revised manuscript and the changes we have made at this position in the revised manuscript.
L 97: Error message instead of a reference
Response: (L 95) We have corrected the error.
L 104-114: Difficult to understand this when one is not familiar with terminology of animal behaviour and training- consider breaking into a few smaller sentences, and defining a few terms?
Response: Modified as below:
(L 102-112) “An ideal outcome for the study would be initiation of movement to the latrine, firstly, in response to the externally-applied stimulus (vibration) and subsequently to other behaviours occurring at the time of the initiation of urination (e.g. internal stimuli associated with a full bladder). There is ample evidence in the literature from rats, pigeons and humans that initiation of behaviours can be readily transferred from one external stimulus to a novel one when the new signal occurs shortly before the first-learned stimulus (e.g., [34-36]). To our knowledge, transfer of control from an exteroceptive stimulus to an internal reflexive excretory signal in cattle has not previously been studied. Interestingly, evidence for the transfer of control between stimuli is provided by two different types of behavioural responses. In one case, experimental subjects orient towards the novel stimulus before the first-learned signal is presented [35], or they may orientate towards the reward location [37].”
L 192: I'd like to suggest a few extra lines at the start of the Methods section, giving a broad overall structure of the Method: defining what is a 'session', a 'trial', a 'part' and a 'phase', and what each step will achieve- maybe around line 192.
Response: We have amended the Methods as follows in line 197-202: “The training was divided into three main Parts (training voluntary behaviours inside and outside of the latrine, and training reflexive responses). Each Part consisted of two or more Phases. The animals were trained once daily (called a session). During a session there were several trials, each beginning with the delivery of an exteroceptive stimulus (audible/vibration signal) and ending with reward delivery, or, in cases where there was no reaction to the signal, 10 s after the signal had been activated.” The information about achievements is already included in the text in L 216-217, L 224-226, L 234-235, L 244-245, L 254-255.
L 209-215: Defining a "Trial" could either come at the very start of this section, before you start describing phases, or may be better in the next section, after Line 303, when you start talking about coding events for each trial.
Response: (L 199-202) The definition of a trial is now presented earlier (at the beginning to the experimental procedures section).
Line 308: Why were certain events recorded for only certain phases? An extra line to explain this may help.
Response: We added the following sentences: (L 317-320) “The reason for this was that in Part 3, no trials were defined that started with an exteroceptive stimulus, instead trials started with the intention to urinate. Similarly, events a) and b) were not coded in Part 3. To take into account the time and duration of confinement in the latrine, event e) was coded in Part 3 only.”
Line 330: Does "and trials" need to be here or is it a typo error?
Response: (L 341) We have deleted this.
Dear reviewer
In the point-to-point reply, we first give the line number in the original manuscript to which the comment refers and the comment itself.
Then we give the line number in the revised manuscript and the changes we have made at this position in the revised manuscript.
L 2-3: Suggest rewording the question to a statement reflecting your results. This will be more informative to the reader. The first part could instead be the question. E.g. "How to toilet train a cow? Incorporating reflexive behaviours into a behavioural chain"
Response: (L 2-3) Reworded analogous to suggestion.
L 21: Suggest rewording "thus" or simply delete "thus far"
Response: (L 21) Deleted as suggested
L 45: Are these 4 words (“reflex conditioning behavioural chain”) all one key word?
Response: (L 45) No, we have inserted a comma after conditioning.
L 58: Suggest rewording to "cows would be less exposed to their excreta"
Response: (L 56) Changed as suggested.
L 65-66: Suggest delete “of course”.
Response: (L 63) Deleted as suggested.
L 70: Suggest rewording to "step-wise"
Response: (L 68) Reworded as suggested.
L 89: Change “so” to "hence"
Response: (L 90) Changed as suggested.
L 91: Change “are” to "include"
Response: (L 86) Changed as suggested.
L 92-93: Change “we used calves in our study” to "calves were used in our study"
Response: (L 88) Changed as suggested.
L 94-104 reads more like materials/methods to me. I suggest to reword or move to section 2.
Response: We agree that some aspects of this section resembles scientific methodology. However, wording here differs from the Methods section in providing justification for the chosen procedures. Further, as the reviewers’ have mentioned, readers unfamiliar with the animal learning literature may find the procedures somewhat difficult to follow. Hence, we have provided some additional background and justification in the Introduction, as is increasingly common in scientific reporting. Nevertheless, the text (originally in Lines 86-96) has been amended as follows:
Moved text originally in L90-93 i.e. “Several studies have found that learning ability is faster in younger animals than in older animals [27,28]. Other advantages of using younger animals include the ease of handling smaller animals and the possibility of utilizing the learned behaviour over their entire lifetime. For these reasons, calves were used in our study.” (now L 85)
to Line 86 after ‘…. not yet been achieved”.
L 97: error in reference
Response: (L 95) We have corrected the error.
L 104-120: Section reads very detailed in comparison to the above sections. I suggest to reword this section to read a bit "easier" in order to keep the reader who may be novel to the cognitive processes. Please also consider which information is needed in order to postulate your hypotheses and the final aim. You may be able to exclude some lines.
Response: (L 102-112) This section has been amended in response to Reviewer 2’s comments.
L 116-117: Suggest to change this to "Whistance et al."
Response: We used the original endNote Style of MDPI for the citation. Nevertheless, we changed it for better readability (L 114).
L 131: Suggest to delete “in this experiment”
Response: (L 129) Deleted as suggested.
L 137: Which training are you referring to? The halter training or the toilet training? Please specify both.
Response: We changed the sentences in the following manner: (L 134-136) “From the 12th day after arrival at the FBN, each calf was halter trained to make later handling easier. The mean age at the start of latrine training was 91 d.”
L 146: Suggest to change to 0.6 m to be consistent to the remaining measures.
Response: (L 145) Changed as suggested.
L 146: What was the background for choosing black and yellow? Are cows able to perceive yellow as distinctive from e.g. white/other bright colors?
Response: We added a sentence behind: (L 146-147) “The black and yellow tape was chosen because cattle can distinguish yellow particularly well from various shades of grey [39].”
L 154: Was this reward familiar to the calves? And was it a reward which you knew calves were motivated for? Feeding motivation is crucial in your experiment, hence the question.
Response: The calves were introduced to the reward before training (see L 191-192) and we knew from this experience and from practical on-farm experience, that calves are keen on this reward.
L 159: audible to you or to the calves? Which sounds are cows/calves able to detect?
Response: Since the calves reacted to the tone before reward was in the bowl, this was a clear sign of hearing the tone. Furthermore, we added some sentences in L 162-167 about the acoustic and visual abilities of cattle.
L 159-161: Same comment as above - why blue, and were you sure that the calves were able to detect the colors and the sounds?
Response: We added few sentences in L 162-167: “Cattle can hear sounds between a range of 23 Hz – 37 kHz [40]; the auditory signal used in this experiment had a frequency of 970 Hz. Furthermore, cattle have been trained to approach a feed source after an auditory signal [41]. It has been shown that cattle can distinguish yellow, pink, red, violet, blue and green from shades of grey [39]. Thus, it was anticipated that the calves would readily react to the acoustic and visual stimuli and approach the reward bowl, which was subsequently confirmed.”
L 173: Why the grey half?
Response: (L 178) We have removed the grey half.
L 180: Meaning similar birth dates grouped or?
Response: Yes, exactly.
L 181: Suggest changing “training (test) treatment” to "test treatment (trained)"...
Response: (L 186-187) Amended to “training treatment (test)” or the “yoked-control treatment (controls)”.
L 183-184: But that is not random - this would be "predetermined"?
Response: The calves were, indeed, randomly assigned. For clarity, the wording has been amended to: (L 188-190) “Test calves were assigned randomly to either the left or right side of the training arena, and the controls were assigned to the side not occupied by the test animals. Thus, blue, turquoise and brown test calves were trained on the left and black and red on the right.”
L 190: so, pulling the halter and leading the calf to the bowl or?
Response: Yes. (L 195) We inserted “using the halter” after “were directed to the bowl”.
L 193: Suggest changing “that” to "when"?
Response: (L 204) This suggestion would change the meaning of the sentence. We have retained the original wording.
L 198-199: Were these persons visible to the calves?
Response: (L 209-210) Yes, this is shown in Fig. 2.
L 205: Suggest to reword this sentence to avoid the "We"
Response: (L 216) We changed the sentence to: “Part 1 started with magazine training…”.
L 205: "magazine training" is only mentioned this once in the article. Is that an established type of training like e.g. positive reinforcement or classical conditioning?
Response: Yes, this is the usual procedure to familiarise the animal with the reward delivery system (https://dictionary.apa.org/magazine-training).
L 206: Again - please consider to give a reference on cattle auditory abilities to argue that your calves could in fact hear the signals you provided. E.g. Heffner and Heffner 1983 and 1992
Response: As mentioned above, we added some sentences in L 162-167.
L 242: maximum?
Response: Lines 244-252 have been amended as follows:
“In Part 3 (training of urination), the calves were expected to learn to wait in the latrine until the completion of urination before receiving any reward. The duration of urination in cattle is usually less than 30 s [42-44]. Thus, the purpose of this next step (Phase 2.2) was to prepare the calves for a delay between latrine entry and reward delivery. To provide a signal that a reward was imminent, a blue flashing light above the dispenser (S5, Figure 1) was activated for the duration of the delay. The delay duration was increased stepwise from 2 s to 30 s (latencies of 2, 4, 7, 11, 15, 20, 25, 30 over successive days). The calf was required to stand calmly in front of the bowl during the delay; otherwise, the reward was not delivered. All four segments and the latrine were available to the calves in this Phase.”
L 247: Delete “(see Figure 1)” as you mention Figure 1 very frequently
Response: (L 255) Deleted as suggested.
L 263: The authors to look at the literature on cattle visual abilities, specifically in relation to color vision e.g. Soffié et al 1980; Gilbert and Arave 1986; and Riol et al 1989
Response: As mentioned above, we added some sentences in L 162-167.
L 280: Change “toilet” to “latrine”
Response: (L 287) Changed as suggested.
L 294: Very helpful overview
Response: Thank you.
L 299: Please explain the coloring of the boxes (the green and yellow) and it may also be an idea to include the termination correct/incomplete and false in the figure
Response: We added a sentence: (L 306-308) “For consistency, the colour of the latrine (yellow) and reward text boxes (green) were selected to match the related areas in Figure 2.”
In order to keep the figure simple, we do not want to insert any more boxes. The caption describes how we have evaluated each behaviour sequence and the reward text boxes also give an indication of the correctness of the sequence.
L 301: Was this person blind to the treatments?
Response: No, but see L 323-325 for intraobserver reliability.
L 303: So, what is the final sample size per treatment in your analyses?
Response: Five test calves and five control calves. We missed some videos as written in L 311-312. The numbers of sessions are given in L 248-250, L 263-264, L 353-355 and in Table 2.
L 320: what does version refer to here? Is it related to SAS?
Response: (L 332) We have inserted “… which is based on R”.
L 325-326: In all the above analyses or only the latter? Please specify.
Response: We added: (L 336) “In the MIXED procedure, the calf was included…”.
L 330: This is the first time you mention the word "learning criteria". Please specify in the previous sections what the specific learning criteria are, and please be consistent in your wording.
Response: We replaced the wording in the Material&Methods part, so it should be better understandable (L 229-232, L 240-243).
L 330: criteria means several/more; criterion means one. Please change accordingly
Response: (L 341) This refers to the different phases of Part 1, so there are three different criteria.
L 335: change “numbers“ to "number"
Response: (L 346) Changed as suggested.
L 337: change “ their controls” to "the controls"
Response: (L 348) Changed as suggested.
L337: suggest rewording to "numerically increased"
Response: We have deleted the sentence.
L 339: suggest rewording to "control calves' trials"
Response: We have deleted the sentence.
L 352: Are cases and trials the same? It is a bit hard to follow the text and the figure.
Response: We have rewritten the sentence: (L 356-358) “Figure 4 shows that in Phase 2.1, the test calves were outside the latrine at the start of 98% of all trials, while the control calves were in the latrine or with the muzzle in the bowl at the start of 75% of all trials.”
L 354: (a) and (b) are not denoted on the figure?
Response: (L 360) We deleted (a) and (b).
L 354: yoked controls/controls/control calves? You use all three - please be consistent in your wording to not confuse the reader.
Response: In response to Reviewer’s comments at Line 181 and our amendment, we have standardized on the word “control(s)” for the remainder of the manuscript.
L 416: again, does this (“cases”) mean trials?
Response: We reworded the sentence (L 421-422): “For sequences in which urination occurred in the latrine (Seq1+, Seq3+, Seq6+), the calves displayed goal tracking (orienting to the reward dispenser) on 63 ± 14% (mean ± SD) of urinations.“
L 420-421: Yes, so why include this? Suggest to delete.
Response: (L 426-427) We want to include this in the discussion of the occurrence of Seq6+.
L 430: It is a bit hard to follow if a urination event and a sequence is the same?
Response: It is not the same. A urination event started with the intention to urinate and stopped with the end of the urination. A sequence can contain more (two) urination events as shown in Figure 3.
L 447-452: I'm not sure which needs to be changed, but you term your external stimulus both "cue", "signal", and "prompt". Please be consistent as the reader loses track of what you mean.
Response: We have reworded “cue” and “prompt” in the whole manuscript.
L 448: This would have been good to include in the argument for why you chose to use this kind of cue in the introductory sections.
Response: We added this information in L 163-164.
L 454: What that in accordance to what you expected?
L 472: please specify what is meant here
Response: We have rewritten the end of the sentence: (L 478-479) “… without the need for delivering external signals, e.g. vibration”.
L 478-479: So, what is your ideas for future studies?
Response: We added a sentence in L 486-487: “Our aim was not to test impulse control in calves, but these results indicate that calves have a degree of self-control, which would be interesting to explore further.”
L 482: which species?
Response: We have rewritten the sentence: (L 489-491) “The control of voiding by inhibiting (or interrupting) in the wrong place and reinitiating excretory responses in the right place (latrine) is a key element in the successful learning of toilet use in other species, e.g. humans and dogs”.
L 505-507: You don't really use this information. Perhaps it could be an idea to include this low percentage more? What does it mean in comparison to your results?
Response: We have rewritten the sentence and added new sentences: (L 513-518) “The Whistance et al. [26] study provides some additional support for the idea that cattle can respond appropriately to internal pre-voiding cues - in 9% of sequences the heifers moved to the trainer before voiding. The higher rate of responding to reflexive cues in our study (31%) is likely attributable to differences in the training regimes e.g. all correct responses in the latrine were rewarded in our study, whereas this would not have been the case in the Whistance et al. [26] study.”
L 516: please consider if you need to use both voiding and urination throughout the paper. If the same thing is meant by both words i suggest to only use urination. If both words are needed to explain a difference, please explain this more clearly.
Response: If it was possible we changed “voiding” to “urination” in the whole manuscript.
L 522: be careful with this word (“evidence”). I suggest changing to "Our study suggests"
Response: (L 532) We have qualified the word “evidence” by “some”, this is relatively cautious.
L 547-559: Please refine the conclusion to only answer your aim and not include discussion. The last sentence read as a repetition from the introduction. If you wish to include this aspect, please relate it to your results. E.g. "Our results may be beneficial in practice as..." or "In practice, our results may be used as a tool to..." or the like.
Response: We have shortened the Conclusion in L 562-566 as follows: “Our training procedure was a variant of a common approach used for toilet training in children, in which excretory behaviour is interrupted, and its resumption is rewarded [24]. These results showed that calves can be toilet trained by incorporating reflexive voiding (urination) responses into a learned voluntary behavioural chain. In farming practice, cattle use of latrines would help in reducing harmful gaseous emissions and improving the hygiene of living quarters.“
Some information of the conclusions are now attached to the end of the discussion: (L 554-560) “For cattle, toilet training might be facilitated by a modification of the procedure in which the reflexive behaviour is first trained to occur in the correct location (latrine) before adding the voluntary behavioural elements including approach to the latrine. This alternative method potentially increases the strength of the learned association between voiding and toileting location. It has also been used successfully with children [23]. Other possibilities for facilitating training include making the latrine more clearly distinguishable from the alley (e.g., by using a unique colour scheme for the latrine walls and floor) and methods for reducing the time spent in the latrine.”
L 549: I suggest to avoid using this "loaded" word as your sample size is small. Rewording could be "Our study showed that reflexive..."
Response: (L 562-566) We have rewritten the conclusion, so the word “loaded” is not used anymore.

Round 2
Reviewer 3 Report
Dear Authors,
It was a pleasure to read the revised manuscript, and thank you for making all amendments.
I have only one smaller comment for the conclusion:
Line 565-566: I suggest changing this to not replicate the introduction. E.g. to something like “In practice, this knowledge may be used on a larger scale to train cattle to use latrines and if successful, this may reduce harmful gaseous emissions [4] and improve hygiene of living quarters [6,8].”
Congratulations on a great piece of work.
Author Response
Dear reviewer
In the point-to-point reply, we first specify the line number in the original manuscript to which the comment refers, as well as the comment itself.
The answer is indicated in red. We indicate the line number in the revised manuscript and the changes we have made at this point in the revised manuscript.
Reviewer 3:
Line 565-566: I suggest changing this to not replicate the introduction. E.g. to something like “In practice, this knowledge may be used on a larger scale to train cattle to use latrines and if successful, this may reduce harmful gaseous emissions [4] and improve hygiene of living quarters [6,8].”
Response:
L 565-567: We have changed the sentence accordingly.
